# Electrospray Deposition of Polyvinylidene Fluoride (PVDF) Microparticles: Impact of Solvents and Flow Rate

**DOI:** 10.3390/polym14132702

**Published:** 2022-07-01

**Authors:** Akinwunmi Joaquim, Omari Paul, Michael Ibezim, Dewayne Johnson, April Falconer, Ying Wu, Frances Williams, Richard Mu

**Affiliations:** 1Research and Sponsored Programs, Tennessee State University, Nashville, TN 37209, USA; ajoaquim@tnstate.edu (A.J.); opaul@tnstate.edu (O.P.); mibezim@tnstate.edu (M.I.); djohn238@tnstate.edu (D.J.); afalconer@tnstate.edu (A.F.); frwilliams@tnstate.edu (F.W.); 2Department of Food Science, Tennessee State University, Nashville, TN 37209, USA

**Keywords:** drug delivery, electrospray, PVDF, microparticles, piezoelectric

## Abstract

Polymeric microparticles have been shown to have great impacts in the area of drug delivery, biosensing, and tissue engineering. Electrospray technology, which provides a simple yet effective technique in the creation of microparticles, was utilized in this work. In addition, altering the electrospray experimental parameters such as applied voltage, flow rate, collector distance, solvents, and the polymer-solvent mixtures can result in differences in the size and morphology of the produced microparticles. The effects of the flow rate at (0.15, 0.3, 0.45, 0.6, 0.8, and 1 mL/h) and N, N-Dimethylformamide (DMF)/acetone solvent ratios (20:80, 40:60, 60:40, 80:20, 100:0 *v*/*v*) in the production of polyvinylidene fluoride (PVDF) microparticles were studied. Scanning electron microscopy (SEM) was used to observe changes in the morphology of the microparticles, and this revealed that a higher acetone to DMF ratio produces deformed particles, while flow rates at (0.3 and 0.45 mL/h) and a more optimized DMF to acetone solvent ratio (60:40 *v*/*v*) produced uniform spherical particles. We discovered from the Raman spectroscopy results that the electrosprayed PVDF microparticles had an increase in piezoelectric β phase compared to the PVDF pellet used in making the microparticles, which in its original form is α phase dominant and non-piezoelectric.

## 1. Introduction

Polymeric microparticles have garnered a lot of interest in the areas of tissue engineering, biosensing, and more importantly in drug delivery systems [1,2]. As a drug delivery system, microparticles (MPs) can be encapsulated in such a way to shelter the core from the environment, thereby providing benefits such as protecting a part of the body from potential side effects, concealing an unpleasant taste, separating incompatible substances, and providing targeted release of the drug which enhances its potency [3]. Using polymeric MPs such as Poly Lactic-co-Glycolic Acid (PLGA) also provides biocompatibility and biodegradability over conventional MPs [4]. In tissue engineering, stimuli generated from a special class of electroactive polymers such as PVDF are promising to promote tissue regenerative response and the stimulation of cellular activities [5]. This is possible due to its capability to generate electrical signals in response to mechanical stress, thereby promoting cellular adhesion, proliferation, and differentiation [6].

PVDF is a well-known biocompatible, lightweight, mechanically stable, and non-toxic piezoelectric polymer with the added benefits of also being used as a binder [7]. This polymer can be synthesized into at least four major crystalline structures, which are α, δ, γ, and β [2]. The α phase exhibits non-piezoelectric properties, but the desired phase for biomedical applications, biosensors, and actuators is the electroactive β phase, which exhibits the strongest piezoelectric response in comparison with the other three phases. 

Among several methods to fabricate the PVDF material such as drop casting, spin coating, spray drying, and electrospray. Electrospray deposition has been considered a promising technique to get high content of desired β phase due to charges induced during spray as well as fabrication of the material as microparticles or nanoparticles [2,8]. The electrospray process involves a polymer solution in a conductive enough solvent that gets emitted from a nozzle charged with a high voltage source.

During the electrospray process to get microparticles, several parameters related to the solvents and polymer-solvent mixtures play a significant role. The solution properties should be able to meet certain conditions in terms of polymer solubility, viscosity, surface tension, density, and conductivity to get the desired material outcome [2]. Optimizing the solution properties and additional parameters related to the electrospray setup also has a great influence on controlling and determining the microparticle shape and size [9]. Among the solution properties, high solution viscosity and surface tension favor increased polymer chain entanglement, which would lead to fibers instead of microparticles [10]. In order to avoid getting fibers, a series of experimental measurements were carried out and PVDF microparticles were obtained at 5 (% *w*/*v*) in which the polymer solution is less viscous. At this stage, it is in the semi-dilute moderately entangled region, where there is a minimal degree of polymer chain entanglement, but it is sufficient to form dense, solid, and reproducible microparticles [2]. Solubility is also an important criterion, as acetone being a poor solvent for PVDF has been previously reported not to form a polymer solution at room temperature alone; thereby, using a co-solvent system such as DMF/acetone provided a solution to this drawback, as DMF is a good solvent that dissolves the PVDF easily, and acetone helps with the evaporation process [11].

In this work, we report on the fabrication of PVDF microparticles by electrospray deposition, we observed the influence of varying the polymer-solvent ratios and the flow rate experimental parameter. We also achieved a stable spray to get spherical, hollow microspheres and deformed particles, which could serve several biomedical purposes; above all, we discovered that the electrosprayed particles possessed the electroactive β-phase, which is absent in commercial PVDF powders and pellets.

## 2. Experimental Section

### 2.1. Chemicals

PVDF (average molecular weight 275,000 Da) received in pellet form and N, N-Dimethylformamide (DMF, 99.8%) were purchased from Sigma-Aldrich, St. Louis, MO, USA. Acetone (ACE, 99.5%) was purchased from VWR chemicals, Radnor, PA, USA.

### 2.2. Material Synthesis

The PVDF solutions were made by dissolving 5 wt.% of PVDF in DMF/ACE co-solvent system in volume ratios of 20:80, 40:60, 60:40, 80:20 and 100:0 (*v*/*v*). DMF is chosen because it is a good solvent for PVDF, and acetone which has high volatility was added to improve the electrospray process by enhancing the evaporation of the solvents. To ensure complete dissolution of polymer in the solvent, the solutions were sonicated for 2 h at room temperature using a water bath sonicator (model B1500A-MTH, VWR, Radnor, PA, USA).

### 2.3. Electrospray Parameters

Electrospray was conducted by applying a voltage of 8.5 kV with an ES50P-20W HV power supply from Gamma High Voltage Research, the electrosprayed solution was emitted from a steel needle with an inner diameter of 22 gauge. A syringe pump was used to feed the polymer solution using various flow rates of 0.15, 0.3, 0.45, 0.6, 0.8, and 1.0 mL/h. The electrosprayed particles were collected on aluminum foil and placed on the grounded collecting plate at 18.5 cm from the needle tip. Experiments were conducted under ambient conditions of relative humidity ranging between 70–75% and temperatures of 17–18 °C.

### 2.4. Characterizations

The morphology of the microparticles was analyzed using a scanning electron microscope (Zeiss Merlin, Gemini II, Jena, Germany), with an accelerating voltage of 2 kV and probe current of 100 pA. In order to get good imaging, the samples were coated with a thin layer of gold using Cressington Sputter Coater. The “ImageJ” software from the National Institute of Health, Bethesda, MD, USA, was used to obtain the average diameter and distribution of the microparticles, for each study, 100 particles from an SEM image were analyzed. The crystal structure of the PVDF pellet and electrosprayed microparticles were investigated using a 532 nm laser Raman Microscope (Thermo Scientific DXR, Thermo Scientific, Waltham, MA, USA). The Raman measurements were performed with microscope objective lens 50×, laser power of 10 mW, and scanning range between 500 to 1600 cm^−1^ at a resolution of 2.7 cm^−1^.

## 3. Results and Discussion

### 3.1. Effect of Polymer-Solvent Ratio

Figure 1 shows SEM images of the PVDF microparticles prepared by electrospray from a 5 wt.% PVDF in DMF/ACE (20:80, 40:60, 60:40, 80:20, 100:0 *v*/*v*) solvent mixture, using 0.3 mL/h flow rate and the resulting microparticle size distribution are displayed in the inset of the figure. Microparticles electrosprayed from (DMF/ACE 20:80 *v*/*v*) solution exhibited a deformed/wrinkled hollow structure; this is attributed to the high amount of volatile ACE solvent, which promoted the rapid evaporation of the co-solvents and fast polymer crystallization before the microparticles landed in the collector, as it has been shown that solvent evaporation and polymer diffusion during the flight of the solution droplet to the collector have a great influence on the final morphology of the electrosprayed particle [12].

On the other hand, microparticles sprayed from (DMF/ACE 40:60 *v*/*v*) solution exhibited a predominantly hollow microsphere structure; this can be attributed to the introduction of more DMF and the reduced ACE solvent, which slows down the evaporation rate of the co-solvents, thus giving the polymer chains sufficient time to homogenize during flight time of the droplet to the collector. However, this solvent ratio is not optimized enough to form microparticles with dense spherical structures.

Microparticles electrosprayed from solution mixture (DMF/ACE 60:40 *v*/*v*) exhibited a spherical geometry with fairly uniform size distribution and smooth surface. This is also related to a rule of thumb proposed by Yao et al. [13], who proposed that if morphologically smooth spherical particles are desired, the modified Peclet number value (Pe) should be kept low. This Peclet number value explains that the competition between solvent-molecule evaporation and polymer diffusion in the droplet determines the shape of a particle. To therefore achieve low values of Pe, the evaporation rate should be reduced, or the diffusion rate of the polymer inside the solution droplet should be increased [13,14]. Although the Peclet number analogy was used qualitatively in this experiment, it was shown to predict particle morphology with accuracy, as the morphology of the fabricated microparticles became more spherical as the evaporation rate of the co-solvents was reduced by the introduction of higher amounts of DMF solvent, which is less volatile. 

Microparticles sprayed from (DMF/ACE 80:20 *v*/*v*) and (DMF/ACE 100:0 *v*/*v*) were observed to be spherical with much larger particle size distributions. With the reduction of the volatile ACE solvent in (DMF/ACE 80:20 *v*/*v*) and its absence in (DMF/ACE 100:0 *v*/*v*) solution, there is a higher tendency for wetting to occur due to incomplete solvent evaporation before getting to the collector. 

Therefore, understanding and controlling the interactions between the polymer chains and the solvent molecules enables access to controlled surface morphologies. In this experiment, we observed a progression from deformed particles to hollow microspheres and then dense smooth spherical microparticles obtained at (DMF/ACE 60:40 *v*/*v*).

### 3.2. Effect of Flow Rate on Microparticle Formation 

The flow rate determines the amount of polymer solution available for electrospraying. Therefore, the flow rate should first be set in a suitable range to achieve a successful spray. Figure 2 shows the effects of the various flow rates of 0.15, 0.3, 0.45, 0.6, 0.8 and 1.0 mL/h in the production of PVDF microparticles from the electrosprayed (DMF/ACE 60:40) solution mixture. For a low flow rate of 0.15 mL/h, the fabricated microparticles had a mix of ring-shaped and spherical particles with an average particle size of 1.845 μm ± 0.397. This could be due to the combined effect of the electric field struggling to keep a consistent jet due to the low amount of solution getting pumped to the needle tip and the quick evaporation of the solvent before the droplets could form dry PVDF particles. The flow rates at 0.3 and 0.45 mL/h produced spherical particles with average particle sizes of 2.671 μm ± 0.256 and 3.563 μm ± 0.275, respectively. Microparticles formed from 0.6, 0.8, and 1.0 mL/h also produced spherical particles with larger average particle sizes and greater size distributions. This is due to the high flow rate and increases in time for complete evaporation of the co-solvent from the solution to form particles. Figure 3 shows the average particle size distribution for all flow rates and solution mixtures of DMF/ACE (40:60, 60:40, 80:20, and 100:0). It is worthy to note that the flow rate at 1.0 mL/h and (DMF/ACE 100:0) solution mixtures experienced excessive wetting, which prevented particle collection.

### 3.3. Polymer Phase Content

The PVDF chemical structure is composed of –CH_2_-CF_2_- unit repeating itself along the polymer chain and characteristic vibrational modes, which can be used to identify the thermodynamically stable α and electroactive β phases.

The structural difference between α and β phases is usually attributed to Raman peaks associated with the vibrations of CH_2_-rocking modes at 797 cm^−1^ for the α phase and 840 cm^−1^ for the β phase, accompanied by the presence of CF_2_—scissoring modes at 610 cm^−1^ and 510 cm^−1^ for α and β crystalline phases, respectively [15].

Figure 4a shows the Raman spectra of PVDF electrosprayed microparticles; also included are α phase dominant PVDF pellet and β-phase dominant measurement specialties 28 μm film used as reference. To study the phase transformation or evolution of the β phase in the electrosprayed MPs, the intensity ratio of the Raman peaks at 797 cm^−1^ and 840 cm^−1^ were calculated, as shown in Figure 4b [16].

Figure 4b shows that the characteristic bands of the β phase in the microparticles increased in intensity compared to the reference α-phase PVDF pellet, but the commercial PVDF still had the highest β-phase content. This might be due not only to the electric field applied to the films but also to the stretching of the films that provide molecular chain alignment at an elevated temperature allowing for dipole alignment along the electric field [17]. The produced microparticles also had the presence of the α -phase peak at a lower intensity. The results showed that varying the solvent mixture did not influence substantially the amount of β-phase content in the electrosprayed microparticles. DMF/ACE 80:20 and 100:0 (*v*/*v*) had the lowest intensity; this could be due to wetting as particles are not fully dried before getting to the collector and more solvents with particles are being introduced during flight time to the collector thereby creating an unstable system. We observe a new peak at 1131 cm^−1^ and 1528 cm^−1^ in these two solvent mixtures, the 1131 cm^−1^ peak has been previously reported and was attributed to a mixture of both α and β phases [18]. We assume based on experimental findings that the signal at 1528 cm^−1^ could be contributed by the mixture of α and β phases, and upon further analysis, in the intensity ratio of β to α phase of the samples, they were present, compared to the electrosprayed microparticles, in which they were absent.

## 4. Conclusions

In summary, PVDF microparticles have been successfully fabricated using the electrospray system. It was observed that varying the ratios of the DMF/ACE co-solvent and flow rate provided an effective system of controlling the size and surface morphologies of the microparticles produced. Microparticles electrosprayed from 5 wt.% (DMF/ACE 60:40 *v*/*v*) using flow rates of 0.3 and 0.45 mL/h produced spherical and fairly uniform particles with average diameters of 2.671 μm ± 0.256 and 3.563 μm ± 0.275, respectively. Raman spectroscopy revealed that electrospray deposition promoted the fabrication of microparticles in the electroactive β-phase, unlike commercial PVDF powders or pellets. This current study presents an electroactive material that exhibits properties that can further be utilized in biomedical applications such as drug delivery and tissue engineering.

## Figures and Tables

**Figure 1 polymers-14-02702-f001:**
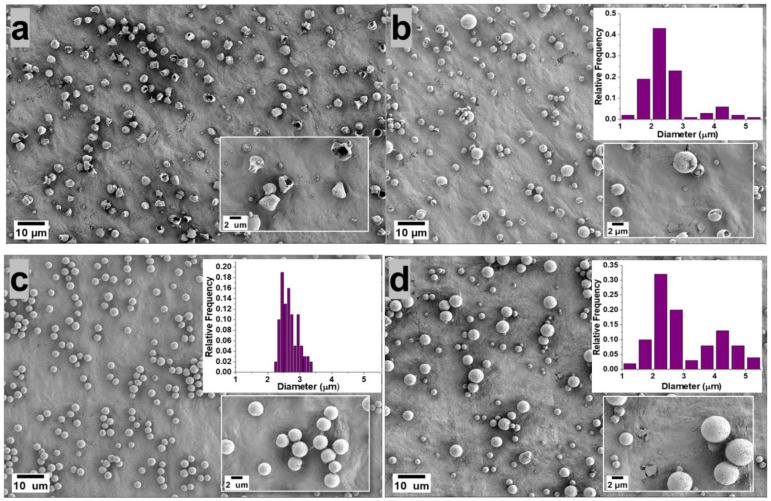
Morphology of the microparticles obtained from 5 wt.% PVDF in DMF/ACE 20:80 (**a**), 40:60 (**b**), 60:40 (**c**), 80:20 (**d**), 100:0 (**e**) polymer solution (*v*/*v*) using flow rate of 0.3 mL/h. The microparticle distribution obtained from each processing condition is presented as an inset.

**Figure 2 polymers-14-02702-f002:**
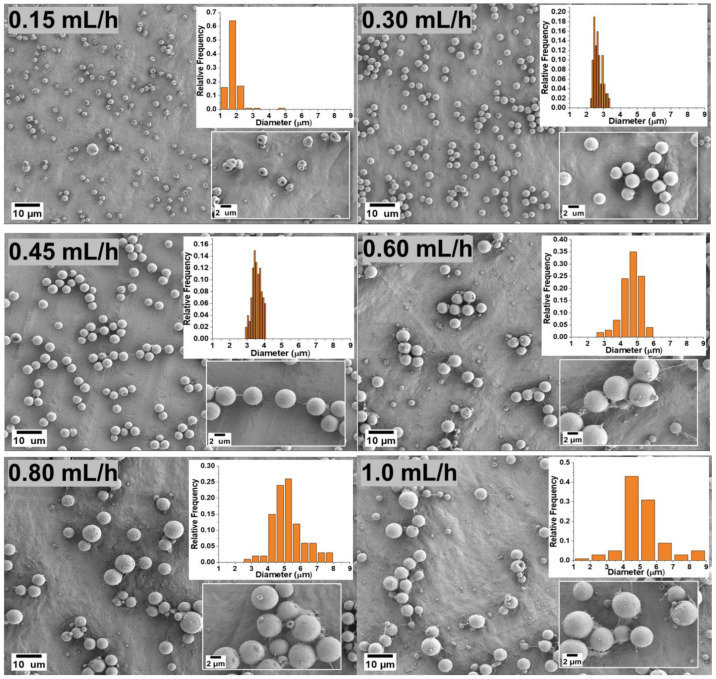
Morphology of the microparticles for the samples obtained with a 5 wt.% PVDF in DMF/ACE 60:40 (*v*/*v*) polymer solution at different flow rates.

**Figure 3 polymers-14-02702-f003:**
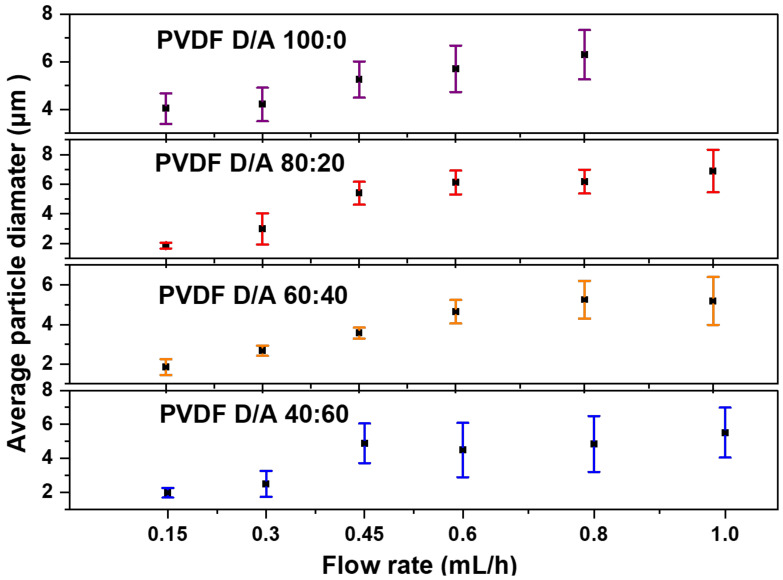
Average PVDF microparticle size distribution of electrosprayed samples varying the flow rate and DMF/ACE solvent ratios.

**Figure 4 polymers-14-02702-f004:**
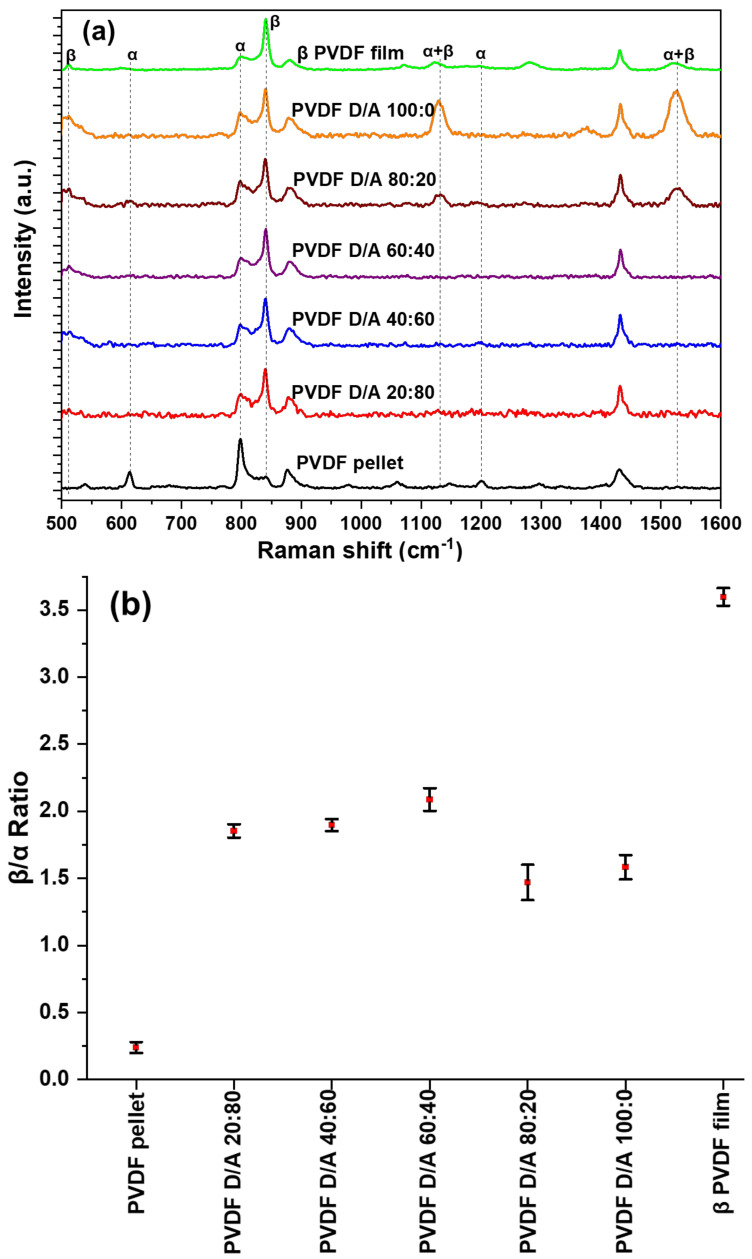
(**a**). Raman spectrums of electrosprayed microparticles, PVDF pellet, and β-phase PVDF film (**b**) Average beta to alpha ratios of samples.

## Data Availability

The data presented in this study are available on request from the corresponding author.

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
