# Peer review of "Electrospray Deposition of Polyvinylidene Fluoride (PVDF) Microparticles: Impact of Solvents and Flow Rate"

_polymers, 2022, doi:10.3390/polym14132702_

Round 1

Reviewer 1 Report

In this work, PVDF microparticles were fabricated through electrospray. The variation of ratio of DMF/ACE and the flow rate can impact the size of the size of the fabricated microparticles and the surface morphologies. The different parameters of fabrication lead to the different size of the microparticles. However, the significance of the size of the microparticles are not elaborated. The authors claim such work can be utilized for biomedical and biosensing applications, which hasn’t been demonstrated in this work. Due to such critical issues, I wouldn’t recommend this work for further consideration.

Author Response

Thank you for taking your time to review this manuscript. We really appreciate your comments and inputs. We have addressed all the comments and concerns in the revised manuscript for your perusal. 

Reviewer 2 Report

The manuscript “Electrospray Deposition of Polyvinylidene fluoride (PVDF) Microparticles: Impact of Solvents and Flow Rate” by Joaquim et al. reports the study of some experimental parameters (mainly solvents mixtures composition and flow rates) and conditions that influence the obtention of PVDF particles in the micrometric range by employing electrospray technology. Despite the topic being very interesting for the field of electrohydrodynamic techniques and microparticles fabrication procedures, I believe authors could cover more aspects related to the characterization of the obtained materials and potential applications in order to enrich the manuscript. My comments and suggestions are attached below:

  1. For a better comprehension of the figures, I recommend reorganizing Figures 1 and 2 of the manuscript, the diameter distributions plots are quite complicated to observe on the SEM images. By the way, why didn't the authors report the relative frequency instead of the number of particles?
  2. Did the authors think about the evaluation of the surface charge for the obtained microparticles? It could be interesting depending on the kind of applications desired for the system in further research.
  3. Why was avoided the evaluation of the changing of other parameters (e.g. distance to the collector or voltage) as the authors suggested in the abstract?
  4. The manuscript could be completed with a report of the conductivity and viscosity of the prepared solutions for electrospraying.
  5. Is noticeable any change in the morphology if the temperature or %RH is altered during the particle fabrication for this particular system?
  6. Figure 4. Authors should label the plots (a, and b, respectively). By the way, I would like to ask the to authors improve the size in the content of plot b.
  7. I am wondering why the authors did not characterize more deeply the PVDF microparticles. E.g. thermal analysis by DSC and TGA techniques or additional spectroscopies techniques could enrich the characterization of the fabricated materials.
  8. Conclusion: “This current study presents an electroactive material that can further be utilized in biomedical and biosensing applications.”. The sentence looks very general taking into account the kind of studies performed in the current version of the manuscript.
  9. The bibliography should be homogenized. I recommend the use of software to help the authors to reorganize the literature citations.
  10. I am curious about the number of authors involved in the research and manuscript preparation: it seems too many authors considering the kind of article submitted and the amount of performed experiments.

Author Response

Thank you for taking your time reviewing this manuscript. We really appreciate the comments and inputs. We have addressed the comments and concerns in the modified version for your perusal.

Reviewer 3 Report

The manuscript is a good contribution to the field of PVDF material.

I have only few questions:

  • Why did you use aluminum foil as a substrate?
  • Why did you provide only Raman measurements for phase analysis?
  • This sentence can be supported by literature: “This polymer can be synthesized into at least four major crystalline structures which are α, δ, γ and β [DOI: 3390/POLYM13152439]“. The small misprints should be corrected, for example, line 103: not “kv”, but “kV”….

Author Response

Thank you for your comments and inputs. We have addressed all the issues in the revised version for your perusal.

Round 2

Reviewer 1 Report

This manuscript can be accepted for publication.

Reviewer 2 Report

The improvement of the manuscript is not enough, I believe the work needs to be completed taking into consideration all the previous suggestions.